# Genome-Wide and Functional View of Proteolytic and Lipolytic Bacteria for Efficient Biogas Production through Enhanced Sewage Sludge Hydrolysis

**DOI:** 10.3390/molecules24142624

**Published:** 2019-07-18

**Authors:** Krzysztof Poszytek, Joanna Karczewska-Golec, Mikolaj Dziurzynski, Olga Stepkowska-Kowalska, Adrian Gorecki, Przemyslaw Decewicz, Lukasz Dziewit, Lukasz Drewniak

**Affiliations:** 1Laboratory of Environmental Pollution Analysis, Faculty of Biology, University of Warsaw, Miecznikowa 1, 02-096 Warsaw, Poland; 2Department of Bacterial Genetics, Institute of Microbiology, Faculty of Biology, University of Warsaw, 02-096 Warsaw, Poland

**Keywords:** sewage sludge hydrolysis, bioaugmentation, whole genome sequencing, metabolic pathway reconstruction, phenomic analysis, anaerobic digestion, biogas

## Abstract

In this study, we used a multifaceted approach to select robust bioaugmentation candidates for enhancing biogas production and to demonstrate the usefulness of a genome-centric approach for strain selection for specific bioaugmentation purposes. We also investigated the influence of the isolation source of bacterial strains on their metabolic potential and their efficiency in enhancing anaerobic digestion. Whole genome sequencing, metabolic pathway reconstruction, and physiological analyses, including phenomics, of phylogenetically diverse strains, *Rummeliibacillus* sp. POC4, *Ochrobactrum* sp. POC9 (both isolated from sewage sludge) and *Brevundimonas* sp. LPMIX5 (isolated from an agricultural biogas plant) showed their diverse enzymatic activities, metabolic versatility and ability to survive under varied growth conditions. All tested strains display proteolytic, lipolytic, cellulolytic, amylolytic, and xylanolytic activities and are able to utilize a wide array of single carbon and energy sources, as well as more complex industrial by-products, such as dairy waste and molasses. The specific enzymatic activity expressed by the three strains studied was related to the type of substrate present in the original isolation source. Bioaugmentation with sewage sludge isolates–POC4 and POC9–was more effective for enhancing biogas production from sewage sludge (22% and 28%, respectively) than an approach based on LPMIX5 strain (biogas production boosted by 7%) that had been isolated from an agricultural biogas plant, where other type of substrate is used.

## 1. Introduction

Anaerobic digestion (AD) of sewage sludge at wastewater treatment plants represents one of the most promising bioenergy production techniques. Its first step—hydrolysis—is usually the rate-limiting step in converting waste substrates into biogas [1]. Diverse approaches, including mechanical, thermal, chemical and biological treatments, were proposed to enhance the process of hydrolysis, and consequently increase, biogas production [2,3]. Broadly defined environment friendliness, minimal formation of inhibitory by-products, low energy requirement, and mild operating conditions [3,4] are the most important advantages in favor of using biological methods over the physical and chemical ones for hydrolysis enhancement. Common biological approaches include bioaugmentation with hydrolytic microorganisms or the addition of hydrolytic enzymes to the system. Whereas the addition of exogenous enzymes to anaerobic bioreactors may boost the performance of AD systems, enzyme activity is affected by a variety of factors, including substrate type, incubation time, system configuration, and the operating conditions (e.g., temperature and pH) [5]. Moreover, the enzymes must often be repeatedly added to a working system, which may render the process more expensive than bioaugmentation with microorganisms.

Bioaugmentation is the practice of adding specific microorganisms or microbial consortia to a system to enhance the desired activity [6]. Microorganisms suited to the bioaugmentation process usually tolerate a wide range of environmental conditions, are capable of growing on unique and diverse substrates by synthesizing unique extracellular degrading enzymes, are robust and competitive after their introduction to a system, and can support the solubilization of organic compounds even in the presence of potential inhibitors [7]. Their ability to produce secondary metabolites, such as vitamins or biosurfactants, may also be exploited in the process. Microorganisms used in bioaugmentation have been isolated from diverse ecological niches, including soil, agricultural residues, manure or animal rumen, where they often form specialized consortia [8]. While microbial consortia may offer a broader degrading potential, they are less biologically stable and controllable than single microbial strains that are hence more often used [9]. The role of bioaugmentation in enhancing methane production was demonstrated for various substrates, which can be grouped into four main categories: sewage sludge [10]; animal manures [11,12]; food industry waste [13,14], and energy crops or agricultural residues [15,16]. These examples are related to studies performed in lab-scale digesters. Bioaugmentation approaches in biogas production could also be grouped according to the metabolic activity (function) that is augmented, i.e., hydrolysis (as addressed in this study), acidogenesis, acetogenesis, or methanogenesis.

The type of substrate used in AD is an important factor affecting AD efficiency, as it influences the composition of microbial consortia and their adaptation to the process in a bioreactor [17]. Sewage sludge in bioreactors is a challenging environment to thrive because it forces the microbes to compete for abundant yet hardly degradable nutrients and to confront various environmental stresses imposed by a working system [7,18]. The survival of microbial strains or consortia introduced to a wastewater treatment plant is the critical factor for the success of any bioaugmentation strategy [7]. It has been observed that under the conditions of industrial processes the bioaugmentation candidate strains often do not express the abilities displayed in lab-scale tests (including enzymatic activities and competitiveness). As a consequence, their amount often decreases shortly after inoculation of a bioreactor [19].

Thus, strain choice has far-reaching consequences for the efficiency of a bioaugmentation process. Whereas remarkable effort was devoted into inoculum strain choice to facilitate biodegradation of sewage sludge [10,20], the availability and attractive cost of whole-genome exploration methods have offered new ways to achieve considerable progress in this topic [21]. Whole-genome sequencing data of environmental isolates provide a valuable groundwork for understanding, predicting and exploiting their metabolic potential in numerous applications, including enhanced biogas production.

The aim of this study was to verify whether and how metabolic potential (encoded in genomic content and expressed under specific conditions) of bacterial strains used for bioaugmentation affects the AD performance. We also investigated the influence of the isolation source of bacterial strains (sewage sludge versus an agricultural biogas plant) on their metabolic potential and their efficiency in enhancing biogas production. In our previous study, we explored a novel bacterial bioaugmentation candidate, *Ochrobactrum* sp. POC9, which had been isolated from a sewage sludge sample [22]. The strain exhibited lipolytic, proteolytic, cellulolytic, and amylolytic activities (confirmed by qualitative tests only) and substantially improved biogas production during anaerobic digestion of sewage sludge. The analysis of the POC9 genome content revealed its denitrifying, biofilm forming, and toxic compound (e.g., phenol) utilization abilities. The conducted genomic and physiological analyses demonstrated that the POC9 strain is resistant to several heavy metals (As(III), As(V), Cd(II), Co(II), Cr(VI), Cu(II), Ni(II), and Zn(II)) and antibiotics, such as β-lactams (including ampicillin, cefexime, cefotaxime, and ceftriaxone), as well as rifampicin and chloramphenicol [22]. In this study, we explored the physiological (metabolic) properties and genome content of two novel isolates–*Rummeliibacillus* sp. POC4 and *Brevundimonas* sp. LPMIX5. We then evaluated their efficiency, as well as the efficiency of a previously studied *Ochrobactrum* sp. POC9, in boosting biogas production from sewage sludge through enhancing the hydrolysis step of anaerobic digestion.

## 2. Results

### 2.1. Physiological Characterization of Rummeliibacillus sp. POC4, Brevundimonas sp. LPMIX5 and Ochrobactrum sp. POC 9

*Ochrobactrum* sp. POC9 and *Rummeliibacillus* sp. POC4 were originally obtained from samples collected from raw sewage sludge at the municipal sewage treatment plant “Czajka” (Warsaw, Poland) while *Brevundimonas* sp. LPMIX5 was originally obtained from a fermenter tank of an agricultural biogas plant (Miedzyrzec Podlaski, Poland). These origin sources differ considerably in various parameters, including the substrate types they provide. The three strains were capable of growing under both aerobic and oxygen-limiting conditions (unpublished data). The three strains were examined and compared for enzymatic activities, optimal culture conditions, metal resistance, as well as the ability to utilize diverse carbon sources and alternative waste substrates.

All three isolates produced hydrolytic zones on tributyrin agar plates (lipolytic activity), Frazier agar plates and nutrient agar plates with skim milk (proteolytic activity) (Table 1). As observed previously [23], a diameter of the hydrolysis zones produced by lipolytic and proteolytic bacteria on solid media may not accurately reflect their enzymatic activities. For this reason, following the preliminary analysis, protease and lipase activities were colorimetrically quantified using a modified Anson’s method [24,25] and a method developed by Gupta et al. [26]. Proteolytic and lipolytic activities are crucial to the hydrolysis step in biodegradation of sewage sludge because proteins and lipids are its main components [1]. Whereas POC4 and POC9 displayed high protease activity (0.49 IU/mL and 0.39 IU/mL, respectively), protease activity in LPMIX5 was 0.10 IU/mL. As one example of previous results of the same protease activity assay, Mongkolthanaruk and Dharmsthiti [27] reported a protease activity of 0.928 U/mL for *Bacillus subtilis* sp. B304–a robust protease producer and a candidate for use in wastewater treatment. Lipase activity determined for LPMIX5, POC9, and POC4 was 3.55, 10.15, and 23.83 IU/mL, respectively. Previously, using the same assay, lipase activity of 18 U/mL was demonstrated for *Halomonas* sp. BRI 8—a psychrotrophic candidate for industrial applications—grown in a medium with olive oil (1%) as a carbon source [28]. Caution should, however, be taken in comparisons of enzyme activity data from various reports. One limitation is that the strains were cultivated under different growth conditions.

A broader biochemical screening for additional enzymatic activities (i.e., cellulase, amylase, xylanase, galactosidase, glucanase, peroxidase, dioxygenase, laccase) was then performed. In these assays, POC4 and POC9 showed cellulase and amylase activities and LPMIX5 showed cellulase and xylanase activities, all crucial to the hydrolysis step of AD (Table 1).

Optimal growth conditions were determined for the three strains. Analyses performed on Lysogeny Broth (LB) medium showed that POC4 tolerated the broadest range of temperatures for growth (10–50 °C). The POC9 and LPMIX5 strains were capable of growing at 15–37 °C (with an optimum at 20 °C) and 10–37 °C (with an optimum at 30 °C), respectively. With the optimal growth at 42 °C, the POC4 isolate was the only moderate thermophile among the tested strains. The fastest growth was observed at pH 7–8, pH 5–7, and pH 6–10 for POC4, POC9, and LPMIX5, respectively (Table 1).

The metal resistance profile of POC9 was determined previously [22] with the use of MIC tests. Here, such an analysis was carried out for POC4 and LPMIX5 strains. The minimal inhibitory concentrations of metals in the analyzed strains were between 1 and 10 mM (Table 1). Such heavy metal (multi-)resistance profiles enable active growth of the strains in challenging environments that are rich in heavy metals, such as wastewater treatment plants.

To shed more light on the metabolic repertoire of the three strains and their potential role in the decomposition of organic matter present in sewage sludge, we performed phenomic analyses with the BIOLOG^TM^ System. The BIOLOG™ EcoPlates allowed to explore the ability of the strains to utilize various carbon sources, including the representatives of a wide array of amines, amino acids, carboxylic and ketonic acids, carbohydrates and polymers, which are all commonly found in sewage sludge [29]. POC4 showed the widest metabolic potential and was able to utilize the highest number of carbon sources (26 substrates). The POC9 and LPMIX5 strains utilized 22 and 13 substrates, respectively (Table 2).

Twelve carbon sources were utilized by all tested strains (i.e., β-methyl-d-glucoside, pyruvic acid methyl ester, d-xylose, l-asparagine, Tween 40, Tween 80, d-mannitol, *N*-acetyl-d-glucosamine, l-threonine, glycyl-l-glutamic acid, d-cellobiose, and α-ketobutyric acid). In contrast, POC9 was the only strain that utilized ɣ-hydroxybutyric acid and LPMIX5 was the only strain to degrade glycogen. l-arginine, 4-hydroxybenzoic acid, itaconic acid, phenylethylamine, and putrescine were uniquely utilized by POC4 (Table 2).

As we were planning to apply these strains as agents hydrolyzing organic waste materials in biogas production, it was necessary to test the possibility for their cost-effective, large-scale production. Therefore, using a modified minimal medium supplemented with 0.1%, 0.5%, or 1% (*v*/*v*) molasses, dairy waste or malt extract we determined the ability of the three strains to grow on alternative waste substrates, which are common by-products of other, parallel processes in biogas plants. The most intensive growth of all tested strains was observed in the medium containing 1% dairy waste after 48 h (Appendix A). Molasses concentration in the range of 0.1-0.5% did not significantly support the growth of the tested strains, however all strains were growing when the concentration was 1%. These abilities of the three strains studied were also evident under oxygen-limiting conditions (our unpublished data), which supported their potential usefulness in bioaugmentation of anaerobic digestion. In the case of malt extract, only the POC9 strain grew, when the concentration of the substrate reached 1% (Appendix A). The conducted experiment demonstrated that the tested strains are capable of using complex substrates (including industrial by-products) as sufficient carbon and energy sources.

Based on the results of physiological analyses, the strains were subjected to further studies, including an advanced genome characterization and analyses of the efficiency of bioaugmentation of anaerobic digestion.

### 2.2. Genome-Guided Exploration of Rummeliibacillus sp. POC4, Brevundimonas sp. LPMIX5 and Ochrobactrum sp. POC 9 as Bioaugmentation Candidates

Sequencing of the *Brevundimonas* sp. LPMIX5, *Rummeliibacillus* sp. POC4, and *Ochrobactrum* sp. POC9 genomes using Illumina MiSeq platform yielded 1,455,743, 545,453, and 1,275,451 paired-end reads and approximately 874,000,000, 327,200,000, and 766,400,000 nucleotides, respectively. The assemblies contained 97, 164, and 255 contigs of a total length of 3,144,717, 3,691,342, and 4,969,575 bp for LPMIX5, POC4, and POC9, respectively. Detailed assembly statistics, average nucleotide identity (ANI), and average amino acid identity (AAI) computed using MIGA webservice, are listed in Appendix A. Predicted plasmid content within assemblies, determined using PlasFlow [30], varied significantly in the tested strains (0.07% in the genome of LPMIX5, 1.78% for POC4, up to 7.07% for POC9).

In our previous study, we explored the POC9 genome content [22]. Coding sequences identified within the LPMIX5 and POC4 genomes in this work were blasted against the COG database [31]. Using RPS-BLAST with an E-value threshold of 1e-5, a total of 2,556 and 2,997 coding sequences of LPMIX5 and POC4 genomes, respectively, were classified into appropriate COG functional categories (Figure 1). More than half of all predicted genes in both genomes were assigned COG numbers associated with cellular metabolism. The largest fraction of the predicted genes was assigned to the E category (as encoding proteins associated with amino acid transport and metabolism), followed by P (inorganic ion transport and metabolism) and C (energy production and conservation) categories. A similar observation was made previously for the POC9 strain [22], which indicates that all three strains harbor complex metabolic (including energy conversion) networks and have the ability to degrade or transform a variety of organic and inorganic compounds, as also shown by enzyme assays and phenomics.

In order to reconstruct hydrolysis-relevant metabolic pathways of LPMIX5 and POC4 strains, their coding sequences, extracted using MIGA webserver, were submitted to KEGG Automatic Annotation System (KAAS), as was done previously for the POC9 strain [22]. KAAS gene mapping revealed that each of the three investigated genomes contains genes encoding enzymes responsible for the digestion of complex compounds. Importantly, each of the inspected genomes contained genes encoding enzymes that enable utilization of long-chain fatty acids (starting from palmitoyl-CoA (16C)) via β-oxidation (Figure 2) and of glutathione degradation crucial for amino acid biosynthesis (Figure 2).

As revealed by culture-based experiments in this study, all of the investigated strains display cellulase activity. Therefore, we inspected their genomes for the presence of the corresponding genes. In the case of the POC4 strain, a bglA-like gene (locus tag: D1606_RS06765) encoding 6-phospho-β-glucosidase (which enables utilization of cellobiose 6-phosphate) was found. Exploration of the POC9 genome revealed the presence of a gene encoding a glycoside hydrolase family 2 protein (locus tag: DK867_08100) that may be responsible for cellulose utilization. Glycoside hydrolase family 2 includes enzymes capable of hydrolyzing a glycosidic bond between two or more carbohydrates or between a carbohydrate and a non-carbohydrate moiety [32]. For the LPMIX5 strain, we identified a gene encoding a DUF2155 domain-containing protein (locus tag: D1604_00980), which has been annotated as a cellulase-like protein and thus may be involved in cellulose utilization.

When the strains were tested in vivo for amylase and xylanase activities under laboratory conditions, amylase activity was detected in POC4 and POC9 strains. We speculated that the POC9 strain might exploit the above-mentioned glycoside hydrolase family 2 protein for the utilization of polymeric carbohydrates, such as starch. The POC4 genome contains a gene encoding a putative cyclomaltodextrinase/maltogenic α-amylase/neopullulanase (locus tag: D1606_13275). Although the LPMIX5 strain did not show any amylase activity under the tested laboratory conditions, amino acid sequence comparisons of *Brevundimonas* sp. genera amylases against the predicted LPMIX5 proteome showed that the strain is probably capable of hydrolyzing glycogen to glucose using amylo-α-1,6-glucosidase encoded within its genome (locus tag: D1604_13050). This interesting finding confirms the previously mentioned limitations of laboratory-driven data used as sole decision tool when choosing a bioaugmentation strain for a specific biotechnological application. In the course of functional analyses, it was also shown that LPMIX5 was the only of the three strains tested that exhibited xylanase activity. Its genome inspection identified two genes that could be responsible for xylan (an abundant biopolymer) degradation capabilities, i.e. xynA-like gene encoding endo-1,4-β-xylanase (locus tag: D1604_09110) and a gene encoding a protein annotated as polysaccharide deacetylase domain protein (locus tag: D1604_12065) that contains a COG0726 domain characteristic of xylanases/chitin deacetylases.

To evaluate the peptide degradation potential of the three strains, their predicted proteomes were screened for peptidase homologues using the MEROPS database [33]. As a result, 273, 310 and 335 proteins were classified in the LPMIX5, POC9, and POC4 predicted proteomes, respectively, as potential peptidases. Most of the degradation-related proteins fell into one of the following groups: cysteine-, metallo- or serine-type peptidases, with metalloproteinases being the most numerous group (Figure 3). This result may indicate that protein degradation capabilities of the three strains could be substantially hampered in environments with low metal ion concentrations or in those rich in metal ion chelating agents.

Bacterial strains intended for usage in bioaugmentation should be resistant to diverse toxic substances, and especially heavy metal ions, to ensure degradation efficiency also under challenging conditions. In the first part of the study, the POC4 and LPMIX5 strains were screened for resistance against the following cations: Cd(II), Cr(VI), Cu(II), Zn(II), Ni(III), Pb(III) under laboratory conditions (Appendix A). Building on the analysis previously conducted for *Ochrobactrum* sp. POC9 [22], genomic exploration of POC4 and LPMIX5 strains was next performed. In silico results revealed that the metal resistance gene profiles for the three strains were similar (Appendix A). It seems that the main mechanism of resistance against divalent ions is associated with the fieF-like gene that encodes an iron-efflux transporter (responsible for iron detoxification) and may mediate some resistance against other divalent metal cations, such as Zn(II), Co(II), Cd(II), and Ni(II) [34]. Resistance against Cr(VI) cations appears to be based on its reduction to Cr(III) by two chromate reductases encoded by chrR-like and yieF-like [35] genes identified in the three genomes. Additionally, the POC4 and POC9 genomes contain chrA-like genes, encoding chromate efflux pumps [36]. Copper resistance strategy appears to be based on binding and sequestering of an excess of copper cations by CopA protein [37]. A corresponding copA gene was identified in all three genomes. Genomic analyses also showed that LPMIX5 strain is equipped with additional nickel efflux systems. Apart from the above mentioned iron-efflux transporter (fieF), LPMIX5 genome contains cnrA-, cnrB-, cnrC-, and crnH-like genes that encode a membrane-bound protein complex catalyzing energy-dependent efflux of Ni2+ and Co2+ [38] and also a nccYXHCBAN operon [39] that encodes a NCC cation-efflux system. Interestingly, the analysis of the LPMIX5 genome identified also a pbrABC operon, which possibly confers lead resistance [40].

Such combination of genes associated with degradation capabilities and heavy metal resistance strategies in the genomes of the three strains studied places the investigated bacteria at the forefront of bioaugmentation-ready strains.

### 2.3. Bioaugmentation of Anaerobic Digestion with Rummeliibacillus sp. POC4, Brevundimonas sp. LPMIX5 and Ochrobactrum sp. POC 9 for Enhanced Biogas Production

Results of genome exploration and physiological tests provided a groundwork to anticipate efficient utilization of complex natural compounds by the three strains studied. Therefore, the effectiveness of degradation of a complex, industrially-relevant substrate–sewage sludge–was next investigated for POC9 [22] and–in this study–for POC4 and LPMIX5. The effect of bioaugmentation on the efficiency of biogas production (through hydrolysis enhancement) was tested in batch scale. Biogas yields were determined daily, while methane content, VFAs concentration and sCOD were quantified after three and seven days, and at the end of the entire process (30 days).

Individual cumulative biogas yields obtained from the batch experiments performed with POC9, POC4 and LPMIX5 reached 294.58, 279.98, and 245.87 dm^3^/kg of VS, respectively (Table 3). In the control variant (sewage sludge without bioaugmentation), cumulative biogas yield was 229.58 dm^3^/kg of VS (Table 3). Thus, bioaugmentation with POC9, POC4, and LPMIX5 increased biogas production from sewage sludge by 28, 22, and 7%, respectively.

The highest increase in biogas production was obtained with the POC9 and POC4 strains, which was presumably a result of the increased efficiency of hydrolysis of organic compounds in sewage sludge in the presence of these robust protease and lipase producers. The highest concentrations of methane in biogas, in variants bioaugmented with POC9 and LPMIX5 and in the control variant, were observed at day 7, reaching 66.48%, 65.91% and 61.34%, respectively. Then methane content decreased in these variants. In the variant bioaugmented with POC4, methane content in biogas was 65.08% after seven days and reached its highest level of 66.88% after 30 days (Table 3).

The chemical parameters such as sCOD and VFAs concentration have been commonly used as indicators of the anaerobic digestion process condition. For the three strains studied in batch assays, VFAs concentration and sCOD were highest after three days. The parameters decreased after seven days and–further–at the end of the experiment (Table 3). After three days, VFAs concentrations were higher by 27%, 16%, and 11% for POC9, POC4, and LPMIX5 variants, respectively, compared to the non-bioaugmented control. In bioaugmented variants, VFAs concentration increased from 1.97 g/L at day 0 to 3.28 g/L (increase by 66.5%), 2.98 g/L (51%), and 2.87 g/L (46%) by day 3 for POC9, POC4, and LPMIX5 variants, respectively. In the control variant, VFAs concentration increased from 1.97 g/L to 2.58 g/L (increase by 31%) over the same time scale. At the end of the experiment, concentrations of VFAs decreased to 0.89 g/L (70% reduction in comparison to day 3), 0.82 g/L (75% reduction), 0.77 g/L (73% reduction), and 0.95 g/L (63% reduction) for POC4, POC9, LPMIX5, and the control (non-bioaugmented) variant, respectively.

In bioaugmented variants, the initial sCOD (5.03 g/L) increased to its highest level in the experiment after three days, reaching 6.97 g/L (POC9), 6.85 g/L (POC4), and 6.70 g/L (LPMIX5). Compared to the non-bioaugmented control at day 3, the levels of sCOD in bioaugmented variants at day 3 were higher by 7%, 5%, and 3%, for POC9, POC4, and LPMIX5 strains, respectively, however the differences were not statistically significant (p > 0.05). High sCOD in the cultures suggested that large amounts of soluble substrates were available for anaerobic digestion. After 30 days, the sCOD values decreased to 3.40 g/L for POC4 (50% reduction compared to day 3), 3.30 g/L for POC9 (53% reduction), and 2.70 g/L for LPMIX5 (60% reduction). In the control variant, sCOD increased from 5.03 g/L to 6.53 g/L after three days and decreased to 3.67 g/L after 30 days (44% reduction vs. day 3) (Table 3).

Obtained results demonstrated that bioaugmentation of anaerobic digestion of sewage sludge with each of the three hydrolytic strains compared (POC4, POC9 and LPMIX5) fosters the efficient hydrolysis of this substrate and, consequently, boosts biogas production. Of the three strains, POC9 and POC4 appear to be the most robust bioaugmentation candidates for biogas production from sewage sludge.

## 3. Discussion

Bioaugmentation with microorganisms may lead to an increase in biomethane output in anaerobic digestion. The literature review shows that, so far, most of the bioaugmentation studies have been carried out on the laboratory scale, yet there is the potential to scale up the process [41]. Various types of microorganisms are required for effective anaerobic digestion, i.e. those involved in hydrolysis, acidogenesis, acetogenesis, and methanogenesis [42]. In this study, we specifically focused on enhancing substrate hydrolysis in an anaerobic digester through the addition of microorganisms capable of breaking down complex constituents of sewage sludge (proteins, lipids, and carbohydrates). We explored the physiological (metabolic) properties and genomic potential of three strains with broad hydrolytic capacities, isolated from two different environments (sewage sludge and agricultural biogas plant) to assess their applicability in bioaugmentation of anaerobic digestion for biogas production.

The results of the physiological analyses (including the BIOLOG^TM^ assay) revealed the wide metabolic versatility of *Rummeliibacillus* sp. POC4, *Ochrobactrum* sp. POC9 and *Brevundimonas* sp. LPMIX5. The bacterial isolates displayed proteolytic, lipolytic, cellulolytic, amylolytic, and xylanolytic activities and were able to utilize diverse carbon sources (amines, amino acids, carboxylic and ketonic acids, carbohydrates, and polymers), including alternative waste substrates (molasses, dairy waste, malt extract) for growth. The results also showed the wide adaptability of the three strains to various growth conditions (temperature 15–37 °C and pH 5–10) and their resistance to diverse metals (Cd, Cr, Cu, Zn, Ni, Pb). With those information at hand survival rates of specific bioaugmentation candidates in an anaerobic digester fed with a specific substrate may be better predicted, and operation conditions may be tailored for improved activity of hydrolysis.

The search for novel microorganisms with exceptional catabolic or survival abilities and potential biotechnology applications is an ongoing pursuit worldwide. Such microorganisms produce specific enzymes in response to the presence of particular substrates in the environment. Many of them are able to utilize peculiar compounds that are not usually preferred by other microorganisms [43]. Sagar and colleagues [44] isolated two lipolytic strains, TU-L1 and TU-L2, from domestic waste dumping site and showed that the optimization of growth conditions, including temperature, pH, agitation (rpm), as well as carbon and nitrogen source had a significant effect on lipase activity. Park and colleagues [45] described morphological and biochemical properties of four proteolytic *Bacillus* strains isolated from a rotating biological contactor in wastewater treatments plants and suggested that these bacteria play an essential role in the degradation of proteinaceous organic compounds in wastewaters. Bramucci and Nagarajan used a combination of traditional microbiological tools and molecular biology for isolation and characterization of 27 different groups or species of bacteria from industrial wastewater bioreactors. The isolates were able to grow on various commercial media and degraded a variety of aromatic compounds (e.g. benzene, toluene, xylene, phenol, cumene). The authors suggested that the microorganisms from wastewater bioreactors are easier to isolate and potentially more amenable to industrial applications than those inhabiting extreme environments [46]. In contrast to those studies, we included detailed analyses of the genome content of the isolated candidate strains for biotechnological applications. The genome-wide study of POC4, POC9 and LPMIX5 strains confirmed the presence of specific genes coding for enzymes enabling the utilization of the following compounds: long-chain fatty acids (starting from palmitoyl-CoA (16C) via beta-oxidation, glutathione, proteins (cysteine-, metallo- or serine-type peptidases), (hemi)cellulose, amylose, and xylose. Results of the genomic analysis also indicated the presence of genes linked to diverse resistance mechanisms to a variety of heavy metals (Cd, Cr, Cu, Zn, Ni, Pb) that are frequently present in sewage sludge. All these genome-driven findings could be useful for tailoring and controlling the sewage sludge degradation process during anaerobic digestion by the strain choice. Previous genome-level studies of microbial biogas producers most often included the phylogenetic and functional characterization of the whole microbial communities inhabiting biogas reactors. In their pioneering effort to deeply characterize the AD microbiome, Campanaro and colleagues [47] identified nearly one million genes and 106 microbial genomes in the biogas microbial community involved in AD. In this work, we first searched for individual, candidate strains for bioaugmentation of AD. We then characterized three such strains through functional assays and genome and metabolic pathway analyses. The genomic data obtained in this study provides a valuable decision tool for developing tailored, site- and substrate-specific bioaugmentation strategies with any of the three strains in any future efforts.

Metabolic versatility is based on the activity of enzymes, especially the hydrolytic (proteolytic, lipolytic, cellulolytic, amylolytic, xylanolytic, etc.) ones. In POC4 and POC9 strains that had been isolated from sewage sludge (which usually contains complexes of proteins, lipids, and a small amount of polysaccharides) only protease, lipase and amylase activities were observed. In contrast, the LPMIX5 strain isolated from an agricultural biogas plant in which maize silage (a lignocellulosic material) is the main substrate, exhibited mainly cellulase and xylanase activities. These findings confirmed the results of previous studies, in which enzymatic activities of isolates were related to the strain’s natural environment and–particularly–to the original substrate. Parawira and co-workers [48] suggested that the nature of a substrate determines the type of the enzymatic activity of the fermentative bacteria present in a digester and observed a higher amylase activity during mesophilic anaerobic digestion of solid potato waste compared to the activities of other hydrolases. Guedon and colleagues [49] showed that cellulolytic *Clostridia* were the dominant strains in anaerobic digesters fed with municipal solid waste or agricultural raw materials containing a high percentage of lignocellulosic compounds. The genomic analysis and functional characterization we performed indicated that POC4 and POC9 are strong candidates for usage in bioaugmentation of anaerobic digestion of sewage sludge, while LPMIX5 may be useful in anaerobic digestion of agricultural waste.

In our studies, POC4, POC9 and LPMIX5 strains were tested for the ability to enhance sewage sludge hydrolysis. These abilities were next verified in the biogas production process from sewage sludge under batch conditions. Results showed that bioaugmentation with POC4, POC9 or LPMIX5 contributes to enhanced biogas production (an increase by 23, 22, and 7%, respectively, compared to a non-bioaugmented control). We also observed an increase in sCOD (by 5, 7, and 3% for POC4, POC9, and LPMIX5, respectively) and VFAs concentration (by 16, 27, and 11% for POC4, POC9, and LPMIX5, respectively) after three days, compared to the non-bioaugmented control. The increased sCOD and VFAs concentration probably resulted from the high metabolic (enzymatic) activity of the strains used for bioaugmentation. The higher efficiencies of biogas production and degradation of organic compounds by POC4 and POC9 strains were probably fostered by the pre-adaptation of these bacteria to the AD substrate (sewage sludge), which originated from the same environment they were isolated from. In our previous work, we evaluated the effect of the isolation source of microorganisms on the selection of hydrolytic microbial consortia dedicated to anaerobic digestion of lignocellulosic biomass [17]. The results indicated that substrate input (and not the community origin) was the driving force responsible for the changes in the community structure of the hydrolytic consortia. In this study, we confirmed the vital role of the type of substrate in the efficiency of anaerobic digestion with specific strains.

Many studies demonstrated the enhanced anaerobic digestion of lignocellulosic biomass through bioaugmentation with enzymes, microbial consortia, or single strains. However, only a few compared the efficiency of diverse single strains isolated from various environmental sources in the augmentation of sewage sludge anaerobic digestion. Lü and co-workers [50] showed that the inoculation with *Coprothermobacter proteolyticus* isolated from a thermophilic digester that was fermenting tannery waste and cattle manure improved the hydrolysis of proteins and polysaccharides and increased methane production by up to 10.7%. Miah and colleagues [51] found that the addition of an anaerobic thermophilic bacterial culture of *Geobacillus* sp. AT1, isolated from aerobically and thermophilically acclimatized sludge, could lead to a 2.1-fold increase in methane production at 65 °C, owing to the protease activity of the strain. Cirne and co-workers [14] showed that bioaugmentation with a *Clostridium lundense* lipolytic strain, isolated from bovine rumen, increased lipid hydrolysis and methane production by 10–20% during anaerobic digestion of restaurant lipid-rich waste. Despite using the same single strain-based approach and a similar substrate, the biogas yield results of the above-mentioned studies are difficult to rigorously compare with those reported here. However, in spite of the varying experimental designs (e.g. differences in bacterial strains, AD temperature, sludge composition, bioreactor size), the biogas yield data may be informative when related to the controls.

As mentioned above, bioaugmentation very often fails. A number of reasons for the failure were suggested, including the growth-limiting conditions due to low substrate concentration; the presence of inhibitory substances (such as antibiotics and heavy metal ions) in the stream to be treated or released by other microorganisms showing antagonistic effects; the presence of bacteriophages; poor biofilm forming ability [52], or adverse operating conditions, such as low temperatures [53]. However, the major assumption is that under the conditions of an industrial process the chosen strains fail to express some of the specific abilities that had been observed in the laboratory with the pure strains following isolation [19]. This variance could be associated with numerous factors, such as the growth rate being lower than the washout rate in reactors, inadequate inoculum size, or insufficient data on the strain ability to use chemical constituents of biomass as sufficient growth substrates. Bioaugmentation failures indicate that lab-scale phenotypical test results are insufficient for making an accurate decision on strain suitability for particular biotechnological application.

Another critical factor for the bioaugmentation success is inoculant compatibility with the indigenous microbial consortium (as indicated previously [54]). Inoculation of a biosystem with a substantial number of cells of a bioaugmentation strain may disturb the system equilibrium affecting the structure and dynamics of the indigenous microbiome (e.g., [55]). In terms of enhancing a desired biodegradation process, these changes may be either beneficial or disadvantageous as they cause system function alteration and shift the reaction equilibriums of the bioprocess [7]. In this study, we focused on enhancing substrate hydrolysis. While indigenous consortium dynamics was not monitored in this study, the bioaugmentation effects reflected as the enhanced biogas production were in line with our expectations.

Previous reports suggested that the addition of nutrients and surfactants or the application of sufficient acclimatization periods may–to some extent–overcome the limitations of bioaugmentation [10]. Based on our study, we suggest that data from genome-wide exploration of the candidate strains may also help prevent bioaugmentation failures in biogas production by providing a better understanding of the degradation pathways, substrate ranges, and survival mechanisms of candidate strains, thus laying the groundwork for an optimal strain choice for specific purposes and conditions. 

Further studies at the level of bacterial transcriptomes and proteomes would broaden the scope of information on the pool of genes that are actually expressed from the genome and on the proteins involved in particular metabolic processes under the specific conditions of an AD process (including the type of the substrate used).

Future work on bioaugmentation of AD of sewage sludge with POC4, POC9 and LPMIX5 strains should involve the determination of the inoculum size that would not affect the biodiversity of the entire bioreactor community, and tests on the washout rate of the added strains after several anaerobic cycles. Tests on a pilot and full industrial scales should also be carried out.

## 4. Materials and Methods 

### 4.1. Bacterial Strains

*Ochrobactrum* sp. POC9 and *Rummeliibacillus* sp. POC4 were isolated from raw sewage sludge collected from the “Czajka” wastewater treatment plant (Warsaw, Poland) as a part of a larger effort. The POC9 strain was previously described in [22]. *Brevundimonas* sp. LPMIX5 was isolated as a part of a larger effort from a fermenter tank of an agricultural biogas plant (Miedzyrzec Podlaski, Poland), where maize silage is used as a substrate. All isolates have been deposited in the bacterial culture collection at the Laboratory of Environmental Pollution Analysis (University of Warsaw, Warsaw, Poland).

### 4.2. Culture Conditions and Enzyme Activity Assays

*Rummeliibacillus* sp. POC4 and *Brevundimonas* sp. LPMIX5 were grown on Lysogeny Broth (LB) [56] at 37°C for 24 h. The medium was solidified by the addition of 1.5% (*w*/*v*) agar. For testing of the proteolytic activity, Frazier agar (BTL, Lodz, Poland) was used. For the assessment of lipolytic and cellulolytic activities, tributyrin agar (Sigma-Aldrich, St. Louis, MO, USA) and CMC-Red Congo agar were used, respectively. For screening of xylanase and amylase activities, soluble chromogenic substrates: AZCL-Xylan (birchwood) and AZCL-Amylose (Megazyme, Bray, Ireland) were added to a nutrient agar medium [22]. In each case, the material from a single colony was transferred onto the specific medium and the plates were incubated at 37°C for 72 h. Visual observation was performed during incubation. The development of clearing zones around bacterial colonies indicated the hydrolysis of a particular substrate. Each experiment was carried out in triplicate. Proteolytic activity was determined by modified Anson’s method [24] using casein as substrate [25]. The method is based on the assessment of the amount of l-tyrosine released during enzymatic hydrolysis of casein carried out by a strain grown on a minimal medium (2 g/L l-asparagine, 7 g/L glucose, 0.96 g/L Na_2_HPO_4_, 0.44 g/L KH_2_PO_4_, 0.2 g/L MgSO_4_ × 7H_2_O, pH 7.0) with 1% casein, and a minimal medium with molasses or dairy waste. After cultivation, the cultures were centrifuged at 6300× *g* for 15 min at 4°C. The supernatant obtained was used as a crude enzyme solution for the estimation of protease activity. The enzyme assay was carried out by incubating 0.1 mL of the enzyme solution with 0.5 mL 0.65% casein in 50 mM phosphate buffer (pH 7.5), pre-warmed to 37 °C, at 37 °C for 10 min. The reaction was terminated by adding 0.5 mL of 110 mM trichloroacetic acid (TCA) and incubating at 37°C for 30 min. After incubation, the reaction mixtures were centrifuged at 13,100× *g* for 5 min. Then, 0.5 mL of 500 mM sodium bicarbonate and 0.1 mL of 0.5 M Folin & Ciocalteu’s reagent (Sigma-Aldrich) were added to 0.2 mL of the supernatant. The reaction was incubated at 37°C for 30 min. l-Tyrosine standards in the range of 0.005–0.2 mg/mL and a blank test (50 mM phosphate buffer, pH 7.5) were prepared as described above. The amount of the released amino acids was quantified spectrophotometrically by measuring the absorbance at 660 nm and using the standard curve. The enzymatic activity of protease was expressed in international units (IU), where one unit of enzymatic activity is defined as the amount of the enzyme that releases 1 μmol of l-tyrosine per ml under the reaction conditions. Lipolytic activity was determined by a modified method developed by Gupta and colleagues [26]. The method is based on the assessment of the amount of paranitrophenol (pNP) generated upon enzymatic hydrolysis of *para*-nitrophenol palmitate (pNPP) in the reaction mixture by the enzymes isolated from the tested microorganisms cultured on a minimal medium supplemented with fats (2 g/L peptone, 1 g/L NH_4_H_2_PO_4_, 0.2 g/L MgSO_4_ × 7H_2_O, 2.5 g/L NaCl, 0.5 mL/L olive oil, 0.5 mL/L Tween 80) and a minimal medium supplemented with molasses or dairy waste. The test was carried out in 96-well plates at room temperature (RT) for 1 h, with absorbance readings at λ = 410 nm every 10 min. Then, 20 μL of a bacterial culture was added to 230 μL of the reaction mixture (3 mg/mL pNPP solution dissolved in isopropanol, 0.9 mg/mL gum arabic solution, 40 mg/mL Triton X-100 solution, 0.5 M Tris HCl buffer, pH 8.8 dissolved in deionized water), in a single well. The enzymatic activity was read from the absorbance curve over time on the basis of the amount of the released pNP. Enzymatic activities of lipase and esterase were defined in international units (IU). One unit of activity was defined as the amount of the enzyme that releases 1 μmol of pNP per mL under the reaction conditions.

### 4.3. Determination of the Optimal Culture Conditions and the Minimal Inhibitory Concentrations of Metals

Bacterial strains were individually cultivated in Lysogeny Broth (LB) medium [56] under a variety of growth conditions: (i) pH 2-12 (set with 2M NaOH or 0.5 M HCl) and (ii) the temperature of 10, 15, 20, 25, 30, 37, 42, and 50 °C.

To determine the minimal inhibitory concentrations (MICs) of heavy metals, LB medium was supplemented with sterile 1-50 mM stock solutions of analytical grade heavy metal salts of cadmium (3 CdSO_4_ × 8 H_2_O), chromium (K_2_CrO_4_), copper (CuSO_4_), lead (PbCl_2_), nickel (NiSO_4_ × 7 H_2_O), and zinc (ZnSO_4_ × 7 H_2_O). The MIC was defined as the lowest concentration of ions that completely inhibited bacterial growth.

To determine the optimal culture conditions and MICs, the cultures were grown for 120 h. A fresh overnight culture was used for inoculation (to the final density of approximately 10^6^ cells/mL). The optical density of the cultures in each variant was determined every 24 h of incubation.

### 4.4. BIOLOG™ Test and Bacterial Growth on Waste Substrates

Carbon metabolism of the three strains was characterized by the community level physiological profiles (CLPPs) using BIOLOG™ Ecoplates (BIOLOG, Hayward, CA, USA) according to the manufacturer’s instructions. Each well of the BIOLOG™ Ecoplate was inoculated with 100 μL of a bacterial culture and incubated under aerobic conditions at constant temperature of 37°C. The plates were scanned at 600 nm with the BIOLOG reader.

The ability to grow on a variety of waste substrates was tested by inoculating a modified minimal medium containing 0.04 g/L NH_4_+, 0.034 g/L NO_3_-, 0.018 g/L P_2_O_5_, 0.02 g/L K_2_O, 0.007 g/L MgO, 0.009 g/L SO_3_, supplemented with 0.1%, 0.5%, 1% (*v*/*v*) molasses or dairy waste or malt extract (Lukasz Drewniak, personal communication). Changes in OD_600_ were measured every 48 h.

### 4.5. Batch Assay of the Anaerobic Digestion of Sewage Sludge

The effect of bioaugmentation of anaerobic digestion by the *Rummeliibacillus* sp. POC4, *Ochrobactrum* sp. POC9 and *Brevundimonas* sp. LPMIX5 strains was investigated in laboratory-scale anaerobic batch experiments. The methanogenic consortium for anaerobic digestion was obtained from a separated fermentation chamber at the “Krym” wastewater treatment plant (Wolomin, Poland). Sewage sludge obtained from the same wastewater treatment plant was used as a substrate in batch assays of anaerobic digestion. Experiments were performed in reactors consisting of 1-L GL 45 glass bottles (Schott Duran, Wertheim, Germany) connected with Dreschel-type scrubbers. To each reactor, a 1-L Tedlar gas bag (Sigma-Aldrich) was attached to collect biogas. The volume of the produced biogas was monitored with the use of MGC-1 Milligascounter (Ritter, Bochum, Germany). Methane content was determined using a gas analyzer GA5000 (Geotech, Coventry, UK). The bioreactors were filled with: (i) the liquid phase from a separated fermentation chamber from wastewater treatment plant, containing the methanogenic consortium inoculate [11 g volatile solids per liter (gvs L-1)], (ii) sewage sludge (11 gvs L-1), and (iii) 4 mL concentrate of POC4 and LPMIX5 (cell density of approx. 10^7^ cells/mL). Anaerobic batch assays were run at 37°C for 30 days in a thermostatic cabinet with magnetic stirring. Physical and chemical analyses were carried out at the beginning of the experiment and after 3, 7, and 30 days. The experiment was performed in triplicate.

### 4.6. Analytical Methods

To monitor anaerobic digestion of sewage sludge, the following parameters were determined: soluble chemical oxygen demand (sCOD), volatile fatty acids (VFAs), and volume and composition of the biogas. The VFAs concentration and sCOD were determined using Nanocolor^®^ kits (Machery-Nagel GmbH, Düren, Germany).

### 4.7. DNA Manipulations and PCR Conditions

Genomic DNA was extracted from bacterial cells using Genomic Mini purification kit (A&A Biotechnology, Gdynia, Poland). The 16S rRNA gene fragment was amplified by PCR with universal primers 27f and 1492r [57]. The amplified fragment was used as a template for DNA sequencing with ABI3730xl DNA Analyzer (Applied Biosystems, Thermo Fisher Scientific, Foster City, CA, USA) at the DNA Sequencing and Oligonucleotide Synthesis Laboratory, Institute of Biochemistry and Biophysics, Polish Academy of Sciences (IBB PAN).

### 4.8. Draft Genome Sequencing and Assembly

DNA preparation, sequencing and assembly procedures were conducted as described previously [22]. Briefly, DNA has been isolated and further sequenced on an Illumina MiSeq platform. Reads obtained were trimmed and assembled using CutAdapt v 1.9.1 [58] and Newbler De Novo Assembler v3.0 (Roche, Basel, Switzerland), respectively.

### 4.9. Bioinformatics

The LPMIX5, POC4 and POC9 genomes were run through The Microbial Genomes Atlas (MIGA) webserver [59] in order to assess the assembly quality and identify putative 16S rRNA sequences for further classification. Putative coding regions were identified using Prodigal [60] and GeneMarkS [61] software packages in such a way that when a region was flagged by both of the above, the region boundaries from GeneMarkS were given precedence. The COG numbers were assigned to each gene by a local RPS-BLAST search against the COG database (last modified 22 January 2015) with 1e-5 e-value threshold by considering only the best BLAST hits [31]. Metabolic features were identified using KEGG database [62] with additional protein searches performed with TBLASTN and RPSBLAST. Analysis of protein degradation potential was conducted by aligning subfamily sequences from MEROPS database [33] using Clustal Omega web service [63]. The multiple sequence alignments obtained were further converted into Hidden Markov Model profiles and searched against amino acid sequences using HMMER suite, version 3.1b1 [64]. Heavy metal resistance genes were identified by searching predicted coding regions against BacMet Predicted Resistance Genes amino acid sequences [65] using DIAMOND sequence aligner [66]. DIAMOND was run with a “--more-sensitive” flag and an e-value cut-off at 0.001. Hits obtained were further confirmed by aligning them to the nr database using BLASTP.

### 4.10. Nucleotide Sequence Accession Numbers

The 16S rRNA gene sequences were compared with bacterial sequences deposited in the GenBank database using the BLAST algorithm provided by NCBI [67]. The 16S rRNA sequence data obtained in this study were submitted to GenBank and published with the following accession numbers: POC4 (MH412677) and LPMIX5 (MH412685). The whole-genome shotgun projects of *Rummeliibacillus* sp. POC4 and *Brevundimonas* sp. LPMIX5 have been deposited in the NCBI GenBank (https://www.ncbi.nlm.nih.gov/genbank) database under the accession numbers QWUA00000000.1 and QWTZ00000000.1, respectively.

### 4.11. Statistical Analysis

Statistical analysis was performed using R (version 3.4.4) in the RStudio environment (version 1.1.463) [68]. Mann–Whitney U tests were performed using the R stats library.

## 5. Conclusions

In this study, we suggested the usefulness of employing genome and metabolic pathway data for strain choice for increasing the hydrolytic activity in biogas reactors. We also investigated the relationship of the isolation source of three bacterial strains and their metabolic potential as well as their contribution to converting sewage sludge substrates into biogas. Genomic and physiological analyses of phylogenetically diverse strains, *Rummeliibacillus* sp. POC4, *Ochrobactrum* sp. POC9 (both isolated from sewage sludge) and *Brevundimonas* sp. LPMIX5 (isolated from an agricultural biogas plant) showed their diverse enzymatic activities and their ability to survive under varied growth conditions. The specific type of the enzymatic activity (proteolytic, lipolytic, cellulolytic or xylanolytic) expressed by the strains was linked to the type of the substrate present in the original isolation source, as the strains were naturally acclimated to a given substrate type. The substrate used in anaerobic digestion also affected the effectiveness of the process carried out with a specific strain. Bioaugmentation with sewage sludge isolates—POC4 and POC9—was more effective for AD of sewage sludge than an approach based on the LPMIX5 strain that had been isolated from an agricultural biogas plant, where other type of substrate is used.

Genome-wide analyses such as those employed in this work complement physiological tests. Their implementation in biotechnological processes may facilitate the choice of the most suitable candidate strain for a specific bioaugmentation process. Having genome sequencing data for a large strain collection at hand may speed up the development of effective inoculants and help prevent bioaugmentation failure.

## 6. Patents

The findings described herein are the subject of Drewniak L., Poszytek K., Dziewit L., Sklodowska A. Consortium of microorganisms capable of hydrolysis of the proteins and lipids in the sewage sludge and/or contaminated soil, the formulation comprising them, the application of the consortium and method of hydrolysis of proteins, lipids and hardly degradable compounds in sewage sludge and/or organic compounds in soils. Polish patent no. PL413998, 2018.

## Figures and Tables

**Figure 1 molecules-24-02624-f001:**
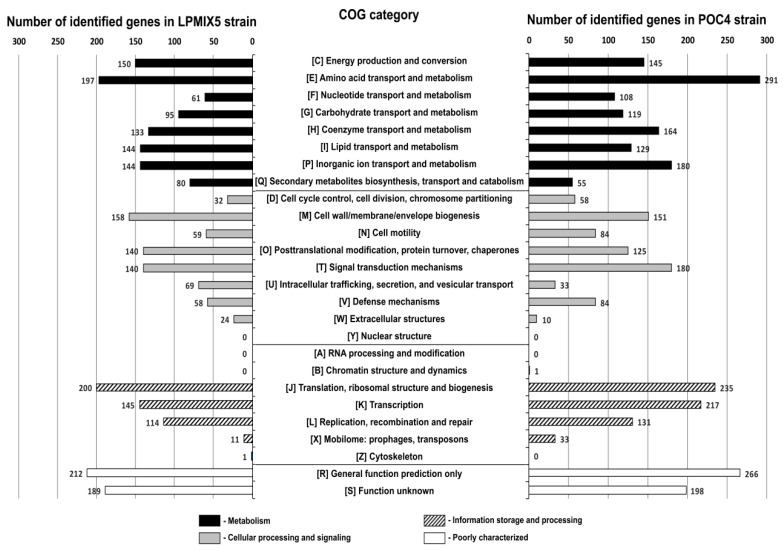
Number of genes of the LPMIX5 and POC4 strains classified into particular COG functional categories.

**Figure 2 molecules-24-02624-f002:**
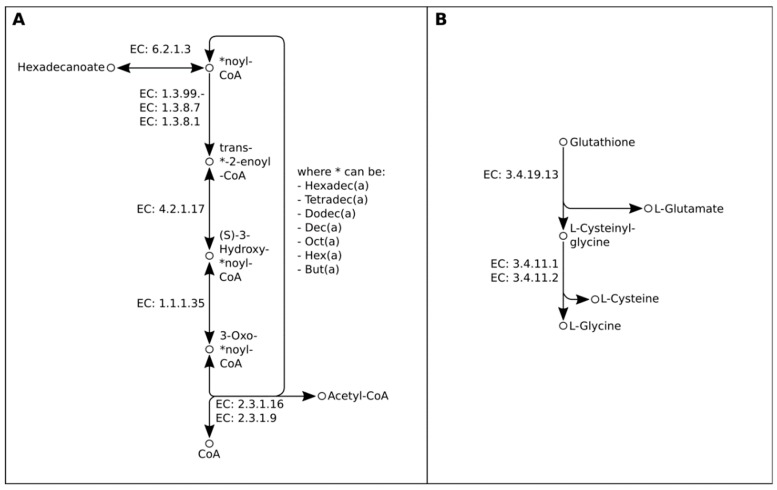
Fatty acid β-oxidation (A) and glutathione (B) degradation pathways of the LPMIX5, POC4 and POC9 strains.

**Figure 3 molecules-24-02624-f003:**
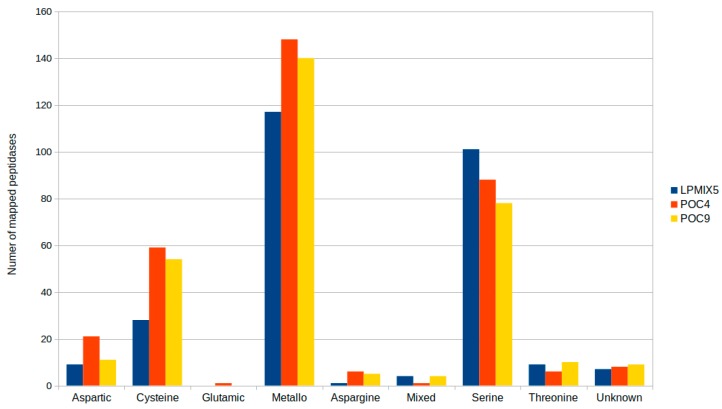
Mapping of the predicted proteins from the three investigated strains to the MEROPS peptidase database clan groups. Aspartic: clans of aspartic peptidases, Cysteine: clans of cysteine peptidases, Glutamic: clans of glutamic peptidases, Metallo: clans of metallo peptidases, Asparagine: clans of asparagine peptide lyases, Mixed: clans of mixed (C, S, T) catalytic type, Serine: clans of serine peptidases, Threonine: clans of threonine peptidases, Unknown: clans of peptidases of unknown catalytic type.

**Table 1 molecules-24-02624-t001:** Optimal culture conditions, minimal inhibitory concentrations of metals, and enzymatic activities determined for the analyzed strains.

Strain	Temperature Range °C	The Optimal Temperature in °C	Growth pH Range	The Optimal pH	The minimal inhibitory Concentrations of Metals (MICs) [mM]						Enzymatic Activity				
					Cd	Cr	Cu	Zn	Ni	Pb	Protease	Lipase	Cellulase	Amylase	Xylanase
*Rummeliibacillus* sp. POC4	7–8				1	5	7.5	5	5	10	+	+	-	+	-
*Ochrobactrum* sp. POC9	5–7				2	10	10	5	7.5	10	+	+	+	+	-
*Brevundimonas* sp. LPMIX5	6–10				2	5	7.5	2	5	10	+	+	+	-	+

**Table 2 molecules-24-02624-t002:** Results of the phenomic analyses.

Carbon Source	Strain		
	POC4	POC9	LPMIX5
β-Methyl-d-glucoside	+	+	+
d-Galactonic acid-ɣ-lactone	+	+	-
l-Arginine	+	-	-
Pyruvic acid methyl ester	+	+	+
d-Xylose	+	+	+
d-Galacturonic acid	+	+	-
l-Asparagine	+	+	+
Tween 40	+	+	+
i-Erythritol	+	+	-
2-Hydroxybenzoic acid	-	-	-
l-Phenylalanine	-	-	-
Tween 80	+	+	+
d-Mannitol	+	+	+
4-Hydroxybenzoic acid	+	-	-
l-Serine	+	+	-
α-Cyclodextrin	-	-	-
*N*-acetyl-d-glucosamine	+	+	+
ɣ-Hydroxybutyric acid	-	+	-
l-Threonine	+	+	+
Glycogen	-	-	+
d-Glucosaminic acid	+	+	-
Itaconic acid	+	-	-
Glycyl-l-glutamic acid	+	+	+
d-Cellobiose	+	+	+
Glucose-1-phosphate	+	+	-
α-Ketobutyric acid	+	+	+
Phenylethylamine	+	-	-
α-d-Lactose	+	+	-
d,l-α-Glycerol phosphate	+	+	-
d-Malic acid	+	+	-
Putrescine	+	-	-

**Table 3 molecules-24-02624-t003:** Physico-chemical characteristics of the anaerobic digestion process. The experiment was performed in three replicates. Standard deviations are included in the table.

Parameters		Cumulative Biogas Production ^1^	CH_4_ Content	VFAs ^2^	sCOD ^3^
**Units**		L/kg_vs_	%	g/L	g/L
	0 day			1.97 ± 0.09	5.03 ± 0.06
Control	3 days	229.58 ± 13.92	43.41	2.58 ± 0.07	6.53 ± 0.12
	7 days		61.34	2.15 ± 0.79	5.83 ± 0.95
	30 days		49.18	0.95 ± 0.09	3.67 ± 0.12
POC4	3 days	279.98 ± 13.58	47.45	2.98 ± 0.77	6.85 ± 1.04
	7 days		65.08	1.60 ± 1.43	5.30 ± 1.89
	30 days		66.88	0.89 ± 0.50	3.40 ± 0.59
POC9	3 days	294.58 ± 44.98	46.00	3.28 ± 0.45	6.97 ± 0.55
	7 days		66.48	1.78 ± 1.38	5.00 ± 2.19
	30 days		58.87	0.82 ± 0.61	3.30 ± 1.62
LPMIX5	3 days	245.87 ± 36.36	46.13	2.87 ± 0.32	6.70 ± 0.53
	7 days		65.91	1.67 ± 0.40	4.93 ± 0.59
	30 days		55.69	0.77 ± 0.25	2.70 ± 0.87

^1^ Cumulative biogas production-total volume of produced biogas converted from organic matter, including the biomass of bacteria. ^2^ VFAs-volatile fatty acids. ^3^ sCOD-soluble chemical oxygen demand.

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
