# Peer review of "Genome-Wide and Functional View of Proteolytic and Lipolytic Bacteria for Efficient Biogas Production through Enhanced Sewage Sludge Hydrolysis"

_molecules, 2019, doi:10.3390/molecules24142624_

Round 1

Reviewer 1 Report

In this manuscript, the authors investigated three strains, POC4, POC9, and LPMIX5, using whole genome sequencing, metabolic pathway reconstruction, and physiological analyses. The results showed that all tested strains display proteolytic, lipolytic, cellulolytic, amylolytic and xylanolytic activities. Bioaugmentation with POC4 and POC9 could enhance biogas production by more than 22% and 28% respectively. There are two minimum comments to authors.

1. In Table 1, the growth temperature range of POC4 is 10-42℃ and the optimal temperature is 42℃. The growth temperature range may be too narrow for POC4. The higher temperature should be tested to find the optimal temperature.

2. Line 326 and Table 3

The change on the levels of sCOD in bioaugmented variants at day 3 is very mild. The authors should calculate their errors to point out its significance.  

Reviewer 2 Report

The paper is really interesting and updated, because it combines different approaches in a constructive way, without neglecting the biotechnological applicative points of view. The introduction is quite exhaustive, as the part dedicated to the methods. In my opinion, however, results and discussions should be restructured. Indeed, in the results section, only these latter should be reported, without comments, references to other works and considerations by the authors, which should be moved to the discussion part. Please revise the manuscript in this key. 

Another point that need to be clarify is the strain POC9, because is not clearly and immediately understandable which results have been produced in this work. Maybe could be useful to produce a table or a short summary about previous results and then ad additional paragraph with new results in comparison with the other two strains. 

Some minor comments:

Lines 111-116: This part sounds like methods rather than results.

Lines 119-126. This section sounds like discussion.

Lines 147-149: It is not necessary to explain what Biolog Plate is.

Lines 175-179: It should be moved in the discussion section.

Line 235: Brevindimonas in italics.

Figure 3: I suggest to use abbreviations in the x axis, instead of single letter; add label to y-axis.

Line 389-390. Please check English language.
